# Investigation on Temperature Behavior for a GaAs E-pHEMT MMIC LNA

**DOI:** 10.3390/mi13071121

**Published:** 2022-07-15

**Authors:** Qian Lin, Lining Jia, Haifeng Wu, Xiaozheng Wang

**Affiliations:** 1College of Physics and Electronic Information Engineer, Qinghai Minzu University, Xining 810007, China; linqian@tju.edu.cn (Q.L.); jln0806@163.com (L.J.); wangxiaozheng666@gmail.com (X.W.); 2Chengdu Ganide Technology Company, Ltd., Chengdu 610073, China

**Keywords:** GaAs E-pHEMT, MMIC LNA, temperature behavior, alpine condition

## Abstract

In order to investigate the temperature behavior for monolithic microwave integrated circuits (MMICs) under alpine conditions, the performance parameters of a 0.4–3.8 GHz gallium arsenide (GaAs) enhancement pseudomorphic high-electron-mobility transistor (E-pHEMT) low-noise amplifier (LNA) are tested at different temperatures. The typical temperatures of −39.2 °C, −32.9 °C, −25.3 °C, −11.3 °C, −4.9 °C, 0 °C and 23 °C are chosen as the alpine condition. The major performance indexes including the direct current (DC) characteristics, S-parameters, stability, radio frequency (RF) output characteristics, output third-order intersection point (OIP3) and noise figure (*NF*), which were inspected and analyzed in detail. The results show that the DC characteristics, small-signal gain (S21), RF output characteristics and *NF* all deteriorate with the rising temperature due to the decrease in two-dimensional electron gas mobility (μ). Contrary to this trend, the special design makes stability and OIP3 increase. For further application of this MMIC LNA under alpine conditions, several measures can be utilized to remedy performance degradation. This paper can provide some significant engineering value for the reliable design of MMICs.

## 1. Introduction

Nowadays, low-noise amplifiers (LNAs), as the key device of radio frequency (RF) receiver front-end, play an extremely important role in aerospace, electronic countermeasures, radar systems, mobile communications and long-distance remote sensing, etc. [1,2,3,4]. Moreover, with the rapid development of RF and 5G technology, development for monolithic microwave integrated circuit (MMIC) LNAs has become the choice of many famous foundries in the world [5], which shows that the investigation of MMIC LNAs is an interesting topic. However, considering the advantages of high electron mobility, high charge density, high power and low noise [6], many parameters of gallium arsenide (GaAs) enhancement pseudomorphic high-electron-mobility transistors (E-pHEMTs) are more sensitive to environmental stress, resulting in a significant increase in failure probability. Meanwhile, studies [7] have shown that the failure of MMICs related to the thermal effect has reached more than 50%, and temperature has become the primary factor affecting the performance of MMICs. When the temperature suddenly changes, the performance degradation for MMIC LNAs is directly related to the quality of the receiving system. Therefore, investigation on temperature behavior for MMIC LNAs is the basic premise to ensure the normal operation of a receiving system. Especially in the Qinghai-Tibet Plateau, the roof of the world, this region has high altitude, large temperature difference and cold climate. The circuit performance and life can be seriously affected due to the sudden change in temperature, which leads to more stringent challenges for MMIC LNAs.

Recently, most of the investigation about MMIC LNAs focuses on the circuit design and relatively little attention is given to its temperature behavior. This means that if its performance degrades to an uncertain level due to the temperature, an irreversible failure for MMIC LNAs with admirable indexes will be caused [8]. According to previous papers, investigation on temperature behavior is still a hot issue [9,10]. However, the common temperature ranges, depending on the extreme temperature, are −40–120 °C, −25–125 °C, 10–90 °C, 20–120 °C or 25–125 °C and so on [8,11,12,13,14,15,16,17,18,19]. Meanwhile, the equal interval temperatures are adopted [8,15,16,17,18,19]. Notably, it is not appropriate to use similar methods to determine the temperature of investigation under alpine conditions. Although there are other ways to study circuit performance [9,20], the research on temperature behavior under certain conditions is absent. Indeed, it is completely necessary to conduct a survey at the actual temperature range. Thus, this investigation on temperature behavior, taking into account the alpine conditions, should be considered.

Here, a 0.4–3.8 GHz GaAs E-pHEMT MMIC LNA is taken as an example to investigate temperature behavior, combined with the actual temperature of the Qinghai-Tibet Plateau from 2018 to 2020. The direct current (DC) characteristics, S-parameters, stability, RF output characteristics and output third-order intersection point (OIP3) for this MMIC LNA are carried out in a temperature range of −39.2–23 °C. Among them, −25.3 °C, −4.9 °C, −32.9 °C and −39.2 °C represent the lowest temperatures in the Qinghai-Tibet Plateau in spring, summer, autumn and winter, respectively. As such, −11.3 °C is the minimum average temperature in the Qinghai-Tibet Plateau. Further, 0 °C is the boundary temperature, and 23 °C represents room temperature for the Qinghai-Tibet Plateau. In addition, the low noise capability for this circuit is specially tested at the highest and lowest temperatures. As a result, the reduction in two-dimensional electron gas mobility (μ) with the rising temperature is the main reason for the degradation of the DC characteristics, small-signal gain (S21), RF output characteristics and noise figure (*NF*). Furthermore, the *NF* also drops when the gate-source capacitance ($C_{\mathrm{gs}}$), source resistance ($R_{s}$) and gate resistance ($R_{g}$) increase. In addition, the stability and OIP3 increase with the rising temperature because of the special circuit design. Finally, in order to further curb the deterioration of performance when temperature suddenly changes, a series of measures can be utilized. All these can provide important references for the reliable design of this MMIC LNA under alpine conditions.

The organization of this paper is as follows: First, the design principle for this MMIC LNA is introduced in Section 2. The experimental design is given in Section 3. Then, the experimental results are discussed in Section 4. Finally, the conclusions are shown in Section 5.

## 2. Implementation of MMIC LNA

In this paper, a GaAs E-pHEMT MMIC LNA is chosen to carry out temperature behavior investigation under alpine conditions. The schematic for this MMIC LNA is shown in Figure 1. In order to ensure the normal operation of the chip, M4 is connected in the power bias circuit to restrain the temperature shift. The gate bias voltage of *V_bias_* is provided by the inductor of L2 and capacitor of C10. The drain power of *V_ds_* is designed with the inductor of L3 and capacitors of C8 and C9. Meanwhile, the capacitors of C1, C2 and transistor of M4 are added to improve the stability for this MMIC LNA. At the output terminal, the capacitors of C7, C9 and resistance of R9 are used to further enhance the stability. Then, the transistor of the M1-based common source structure and inductor of L1 are connected in series to achieve good broadband output matching and high-power characteristics. Moreover, the input impedance matching can be realized only by adjusting L1 when the *NF* changes little. Additionally, the transistors of M2 and M3 with a cascade structure are used in circuit. The influence of the Miller effect on broadband reduces through this structure, which makes this MMIC LNA achieve high gain, wide bandwidth, high linearity and low noise. Finally, the feedback structure is mainly composed of the resistance of R1 and capacitor of C3. This design is aimed to optimize the gain compression, linear distortion and temperature sensitivity of the MMIC LNA. The experiment fixture of this MMIC LNA with Dual Flat Non-lead (DFN) Package is shown in Figure 2. Its area is only 2 mm × 2 mm.

## 3. Experiment Design

The measurement circuits and experimental environment for this MMIC LNA are given in Figure 3 and Figure 4, respectively. During investigation, this MMIC LNA is placed in a temperature test chamber at different temperatures with DC power. The drain-source voltage ($V_{ds}$) and gate-source voltage ($V_{gs}$) are set as 5 V. While measuring S-parameters, stability and RF output characteristics, the vector network analyzer (VNA) ZNB-8 is used. To achieve the maximum output power of this MMIC LNA, it is necessary to add a driver amplifier (DA) at the input terminal. Meanwhile, its output terminal is connected with an attenuator of −30 dB to ensure the safe use of instruments. While testing OIP3 and *NF*, the vector signal generator SMW200A and spectrometer FSW of ROHDE&SCHWARTZ are employed. The dual-tone signals are input through a power combiner (PC) while studying OIP3. Moreover, it is worth noting that the attenuator should be placed in front of the temperature test chamber at the time of measuring *NF*. Finally, the settings of experimental time are shown in Figure 5. In order to better simulate the alpine conditions, after repeated tests and statistics, the ramping and dwell time are set to 60 min and 30 min, respectively. Among them, −25.3 °C, −4.9 °C, −32.9 °C and −39.2 °C can represent the lowest temperatures in the Qinghai-Tibet Plateau in spring, summer, autumn and winter, respectively. As such, −11.3 °C is the minimum average temperature in the Qinghai-Tibet Plateau. Further, 0 °C is the boundary temperature and 23 °C represents room temperature in the Qinghai-Tibet Plateau. Moreover, −40 °C and 105 °C are the lowest and highest temperatures for this MMIC LNA to work normally. Through the above settings, the experimental environment and data are receivable.

## 4. Experiment Results and Discussion

In view of the shrewd temperature variations and cold weather, modules in the communication system would suffer a severe reliability challenge. In order to investigate the temperature behavior of this MMIC LNA in alpine conditions, the typical temperatures of −39.2 °C, −32.9 °C, −25.3 °C, −11.3 °C, −4.9 °C, 0 °C and 23 °C are chosen as the alpine condition. The major performance indexes include the DC characteristics, S-parameters, stability, RF output characteristics, OIP3 and *NF*. The detailed studies and deep analysis are given as follows:

### 4.1. Temperature Behavior on DC Characteristics

The measured curves of the output characteristic and the transfer characteristic for GaAs E-pHEMT with typical alpine temperatures are shown in Figure 6 and Figure 7, respectively. It can be seen that the saturated drain-source current ($I_{ds}$) at room temperature is 56 mA. The $I_{ds}$ reduces from 63 to 56 mA with the rising temperature. Meanwhile, the $I_{ds}$ for GaAs E-pHEMT is expressed by the following equations [8,21]:(1)$$I_{ds}=\{\begin{array}{ll} \mu \frac{\epsilon }{d}\frac{W}{L}V_{ds}(V_{gs}-V_{T}-\frac{1}{2}V_{ds}) & V_{ds}<V_{gs}-V_{T}\\ \frac{1}{2}\mu \frac{\epsilon }{d}\frac{W}{L}{\left(V_{gs}-V_{T}\right)}^{2} & V_{ds}\geq V_{gs}-V_{T} \end{array}$$
(2)$$\mu \propto T^{-3/2}$$
where μ is two-dimensional electron-gas mobility, ϵ is dielectric coefficient, *d* is thickness, W is gate width, L is gate length, $V_{T}$ is the threshold voltage, $V_{gs}$ is the gate-source voltage, $V_{ds}$ is the drain-source voltage and T is the ambient temperature. It can be concluded that the reduction in $I_{ds}$ is related to the decrease in μ [22]. Moreover, $V_{T}$ drops with the rising temperature, resulting in an increase in $I_{ds}$ [23]. However, the reduction trend of $I_{ds}$ is not changed due to the decrease in $V_{T}$ [24]. Thus, the phenomenon shows that as the temperature rises, the key factor causing the degradation for $I_{ds}$ is the reduction in μ.

### 4.2. Temperature Behavior on S-Parameters

The measured curves of S-parameters for this MMIC LNA with typical alpine temperatures are shown in Figure 8. It can be seen that S21 is up to 22.62 dB at room temperature. When the temperature rises from −39.2 to 23 °C, S21 gradually decreases, and the maximum reduction is about 2.03 dB. After deep analysis, it can be found that the key reason for the reduction in S21 is that the trans-conductance ($g_{m}$) decreases by rising temperature, which can lead to the degradation of S21 [25,26]. The $g_{m}$ is expressed as [8,21]:
(3)$$g_{m}=\{\begin{array}{cc} \mu \frac{\epsilon }{d}\frac{W}{L}V_{ds} & V_{ds}<V_{gs}-V_{T}\\ \mu \frac{\epsilon }{d}\frac{W}{L}\left(V_{gs}-V_{T}\right) & V_{ds}\geq V_{gs}-V_{T} \end{array}$$

According to Equation (3), the reduction in $g_{m}$ is also related to the decrease in μ. Finally, S21 degrades significantly. Furthermore, the input return loss (S11) and output return loss (S22) are overlapped in a frequency range of 0.4–4.8 GHz because the circuit impedance mismatches by rising temperature. However, in a frequency range of 0.4–3.8 GHz, S21 is greater than 20 dB, and S11 and S22 are less than −10 dB and −8 dB, respectively. In addition, the maximum value of reverse isolation (S12) is about −29 dB. These results indicate that S-parameters for this MMIC LNA remain stable with the rising temperature.

### 4.3. Temperature Behavior on Stability

The measured curves of stability factor (*K*) for this MMIC LNA with typical alpine temperatures are shown in Figure 9. The expression for *K* is as Equation (4) [21]:(4)$$K=\frac{1-{\left|S_{11}\right|}^{2}-{\left|S_{22}\right|}^{2}+\left|S_{11}S_{22}-S_{12}S_{21}\right|}{2\left|S_{12}\right|\left|S_{21}\right|}$$

It can be seen that an increasing trend for *K* is presented with the rising temperature. The *K* is always greater than 1. This is because the resistance of R9, capacitors of C1, C2, C7, C9 and transistor of M4 are added during circuit design. Finally, the *K* is improved with the rising temperature, which implies that the stability improves.

### 4.4. Temperature Behavior on RF Output Characteristics

The measured curves of RF output characteristics for this MMIC LNA with typical alpine temperatures are shown in Figure 10. It can be seen that the output power (Pout), gain and power-added efficiency (PAE) degrade with the rising temperature. When the frequency is 2.1 GHz, the saturation Pout, saturation gain and maximum PAE at room temperature are 19.23 dB, 20.64 dBm and 28.59%, respectively. Because of the rising temperature, the Pout, gain and PAE in 2.1 GHz drop about 0.42 dB, 0.94 dBm and 3.36%, respectively. After verification and analysis, the main reason for this degradation is that the rising temperature can lead to an increase in the output resistance ($R_{ds}$) [10,24]. Meanwhile, $R_{ds}$ is expressed as [8,27]:
(5)$$R_{ds}=\frac{\partial V_{ds}}{\partial I_{ds}}=\frac{1}{\mu \frac{\epsilon }{d}\frac{W}{L}(V_{gs}-V_{T}-V_{ds})}$$

According to Equation (5), as the temperature rises, $R_{ds}$ increases due to the reduction in μ, which causes more power dissipation. Therefore, the Pout drops. The gain and PAE are as follows [8]: (6)Gain=Pout−Pin
(7)$$\mathrm{PAE}=\frac{\mathrm{Pin}\left(\mathrm{Gain}-1\right)}{P_{\mathrm{dc}}}$$
where P_dc_ is the DC power. It is clear from (6) that as temperature rises, gain also decreases because of the reduction in μ. Moreover, P_dc_ increases after thermal storage [8]. Thus, PAE reduces with the rising temperature according to (7). In a word, the RF output characteristics for this MMIC LNA degrade severely due to the reduction in μ.

### 4.5. Temperature Behavior on OIP3

The measured curves of OIP3 for this MMIC LNA with typical alpine temperatures are shown in Figure 11. In the whole frequency band, its maximum value is 30.93 dBm at room temperature. As the temperature increases, an upward trend is presented for OIP3. Its largest increase is about 3.95 dBm. This phenomenon is mainly related to the circuit design [16]. For this MMIC LNA, OIP3 is better improved and balanced by adding transistor M2, transistor M3, resistance R1 and capacitor C3.

### 4.6. Temperature Behavior on NF

The measured curves of *NF* for this MMIC LNA at different temperatures are shown in Figure 12. When temperature rises from −40 °C to 23 °C, it can be seen that *NF* only deteriorates by 0.2 dB. In other words, the performance of this MMIC LNA is relatively stable under alpine conditions. As the most critical index, the investigation for the lowest and highest temperature is specially carried out to verify the low-noise ability here. Hence, −40 °C and 105 °C are chosen. According to Figure 12, the *NF* at room temperature is only 0.8 dB. Although the *NF* increases with the rising temperature, it still has low-noise performance.

The study of [28] showed that *NF* is expressed by the FUKUI model. The equations are as follows [28]:(8)$$NF=10lg\;F_{min}+\frac{R_{n}}{R_{ss}}\left[\frac{{\left(R_{ss}-R_{opt}\right)}^{2}+{\left(X_{ss}-X_{opt}\right)}^{2}}{R_{opt}^{2}+X_{opt}^{2}}\right]$$
(9)$$F_{min}=1+0.016fC_{gs}\sqrt{\left(R_{s}+R_{g}\right)/g_{m}}$$
(10)$$R_{n}=\frac{0.03}{g_{m}^{2}}$$
(11)$$R_{opt}=2.2\left(\frac{1}{4g_{m}}+R_{s}+R_{g}\right)$$
(12)$$X_{opt}=\frac{160}{fC_{gs}}$$
where $F_{min}$ is the minimum noise factor, $R_{n}$ is the equivalent resistance, $R_{ss}$ is the signal source resistance, $X_{ss}$ is the signal source reactance, $R_{opt}$ is the optimum signal source resistance, $X_{opt}$ is the optimum signal source reactance and *f* is the working frequency. $C_{gs}$ is the gate-source capacitance, $R_{s}$ is the source resistance and $R_{g}$ is the gate resistance. As the temperature rises, $C_{gs}$, $R_{s}$ and $R_{g}$ increase [16]. On the contrary, the $g_{m}$ drops. As a result, $F_{min}$ increases with the rising temperature. Meanwhile, according to the analysis, the *NF* must increase with the rising temperature. The analysis and measured curves are consistent. The increase in *NF* and $F_{min}$ must be related to the rising temperature. Therefore, $C_{gs}$, $R_{s}$, $R_{g}$ and $g_{m}$ are the key factors for *NF* when the temperature rises. Among them, $g_{m}$ drops due to the reduction in μ.

In short, trough the investigation, the life of this MMIC LNA is shortened. Meanwhile, the reasons for performance deterioration were deeply analyzed with the rising temperature. In fact, the deterioration is inevitable. Thus, in order to improve the working time of this MMIC LNA under alpine conditions, some measures should be considered to minimize this deterioration. In the future, the deterioration due to the rising temperature can be cured from the following aspects. On the one hand, for transistors, L is not an ideal designable parameter because the channel modulation effect is significantly affected by its reduction. Further, it is difficult to control L in the process [29]. The degradation due to the reduction in μ can be compensated by increasing W. On the other hand, a temperature compensation circuit with stacked structure suppresses the degradation of S21 and RF output characteristics [30]. Meanwhile, the RF output characteristics are improved by tightly controlling the density of surface states and surface trap levels for GaAs E-pHEMT [31]. Further, when the temperature becomes high, the wires will be short-circuited or broken, which can directly affect the service life for this MMIC LNA. Therefore, the lifetime for this MMIC LNA can be extended by designing a reasonable layout, controlling the process strictly and designing a multi-layer structure [32].

In fact, although temperature is only one of factors that affects circuit performance, more than 50% of circuit failure is caused due to the changes in temperature. The influence of temperature on circuits cannot be ignored. In this investigation, the typical alpine temperatures are selected at the average values of pressure and humidity of a representative city in the Qinghai-Tibet Plateau. However, it is actually a typical temperature experiment. If circuits are required to further adapt to this alpine condition, thermal shock, temperature storage, pressure and humidity must also be considered. According to these factors, the final working range for circuits can be deduced. Once this range is exceeded, a rapid decline in performance of circuits will be caused. The permanent failure for circuits will occur because the degradation is irreversible at this time. Therefore, the degradation of various circuits caused by the above factors will be a hot spot. In the future, a comprehensive investigation should be considered under alpine conditions.

## 5. Conclusions

In this paper, the temperature behavior for a GaAs E-pHEMT MMIC LNA is studied, taking into account the typical alpine temperatures. According to this investigation, as the temperature rises, there are two different phenomena. One is that the DC characteristics, S21, RF output characteristics and *NF* degrade. The other is that stability and OIP3 increase. The results show that the reduction in μ and circuit design for this MMIC LNA are the main reasons for the above phenomenon, respectively. In addition, the *NF* also degrades when $C_{gs}$, $R_{s}$ and $R_{g}$ increase due to the rising temperature. However, this MMIC LNA still has low-noise performance. Therefore, this MMIC LNA can be applied in alpine conditions. Moreover, the paper proposes that, in the future, the performance of this MMIC LNA can be improved by choosing a compromised W, designing a temperature compensation circuit, reducing the density of surface state and surface trap and optimizing layout design. This can provide a significant engineering guide for the reliable design and enhance the core competitiveness of MMICs.

## Figures and Tables

**Figure 1 micromachines-13-01121-f001:**
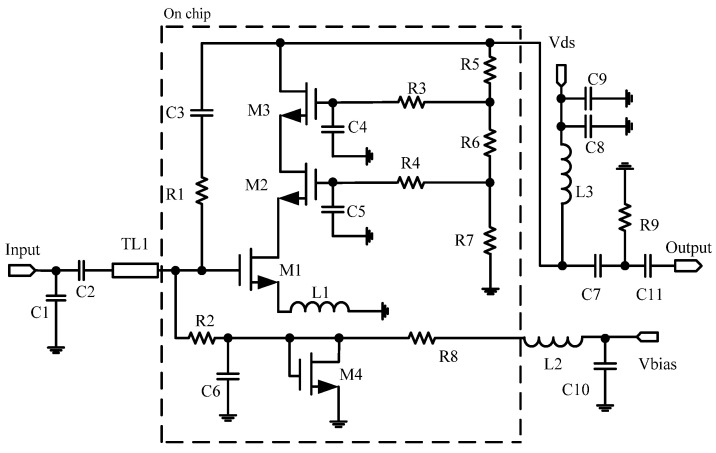
The circuit schematic of the LNA.

**Figure 2 micromachines-13-01121-f002:**
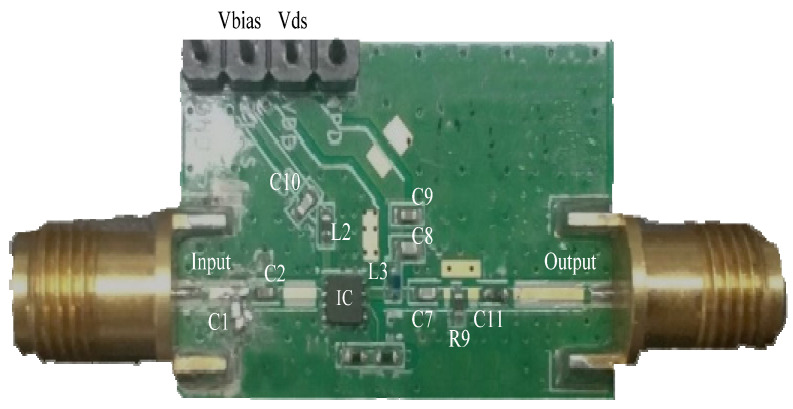
The experiment fixture.

**Figure 3 micromachines-13-01121-f003:**
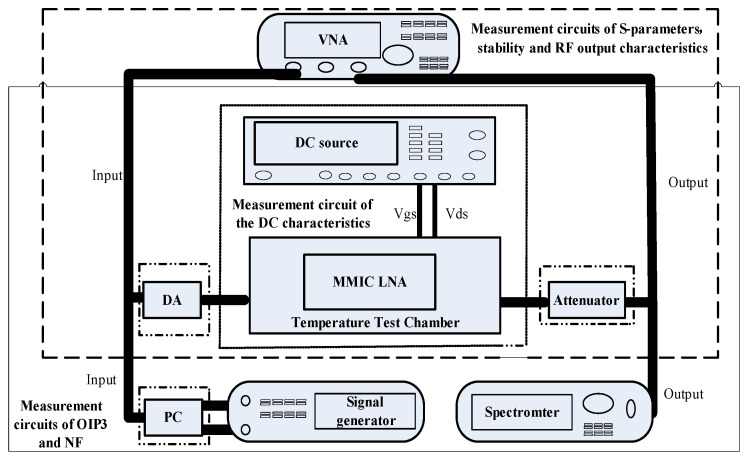
Measurement circuits.

**Figure 4 micromachines-13-01121-f004:**
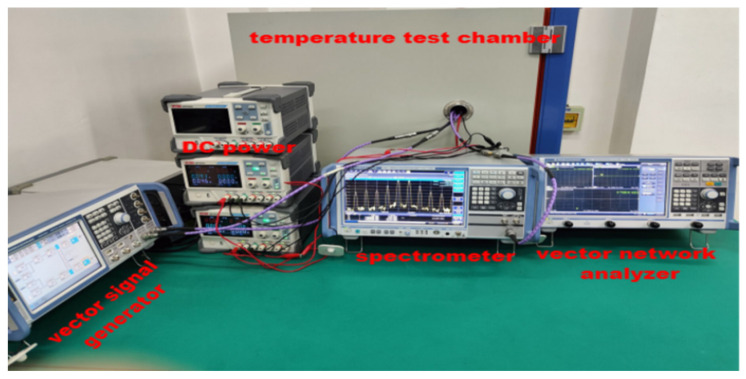
Experimental environment for this MMIC LNA.

**Figure 5 micromachines-13-01121-f005:**
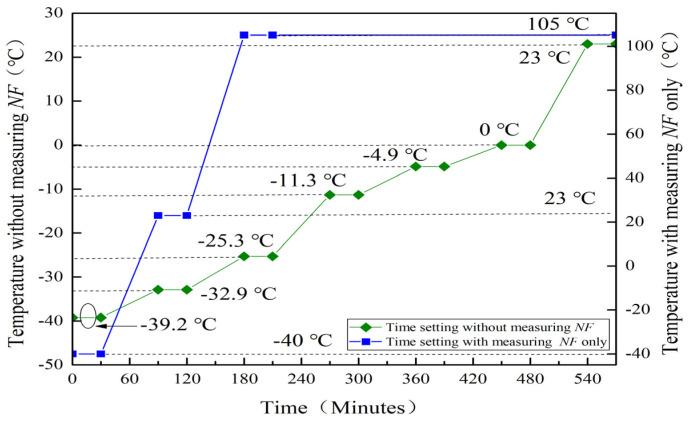
The settings of experimental time for this MMIC LNA.

**Figure 6 micromachines-13-01121-f006:**
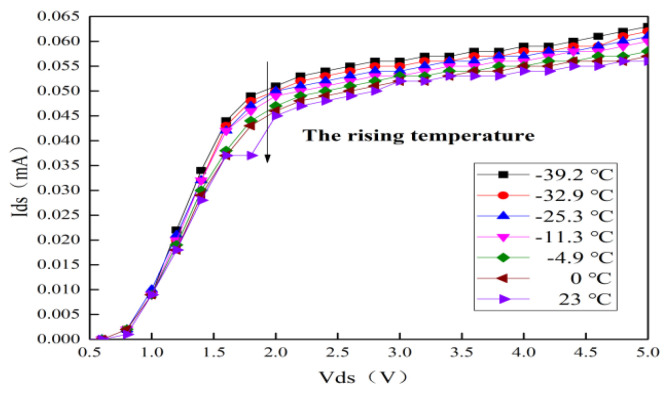
Measured curves of output characteristic with typical alpine temperatures.

**Figure 7 micromachines-13-01121-f007:**
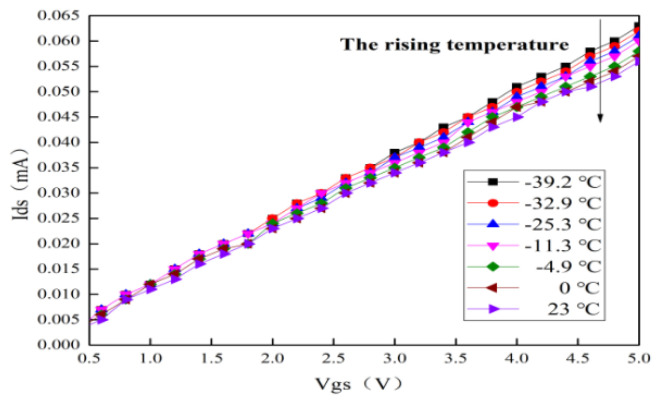
Measured curves of transfer characteristic with typical alpine temperatures.

**Figure 8 micromachines-13-01121-f008:**
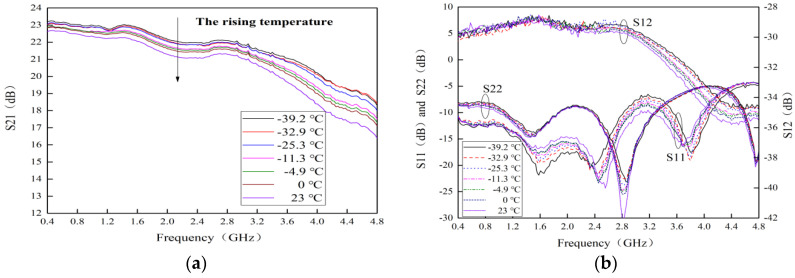
Measured curves of S-parameters with typical alpine temperatures. (**a**) S21 (**b**) S11, S12 and S22.

**Figure 9 micromachines-13-01121-f009:**
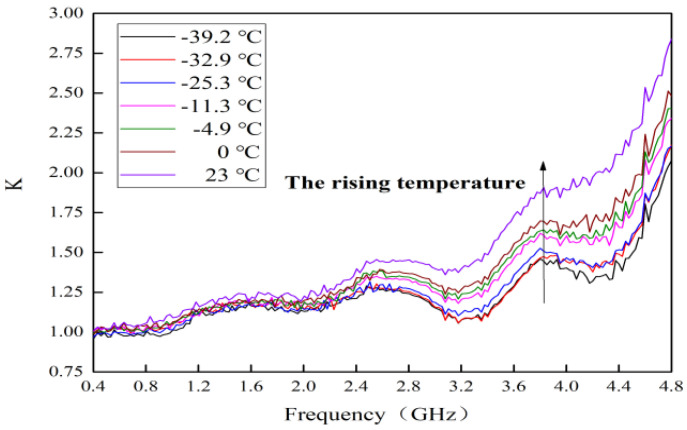
Measured curves of stability factor with typical alpine temperatures.

**Figure 10 micromachines-13-01121-f010:**
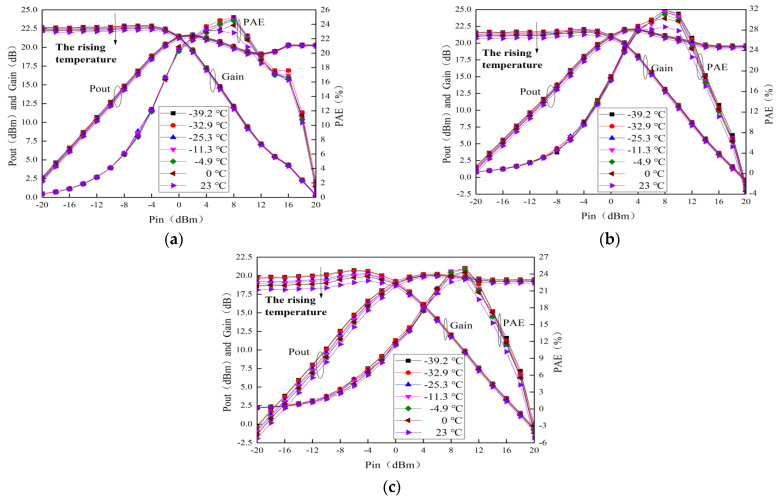
Measured curves of RF output characteristics with typical alpine temperatures. (**a**) 0.4 GHz (**b**) 2.1 GHz (**c**) 3.8 GHz.

**Figure 11 micromachines-13-01121-f011:**
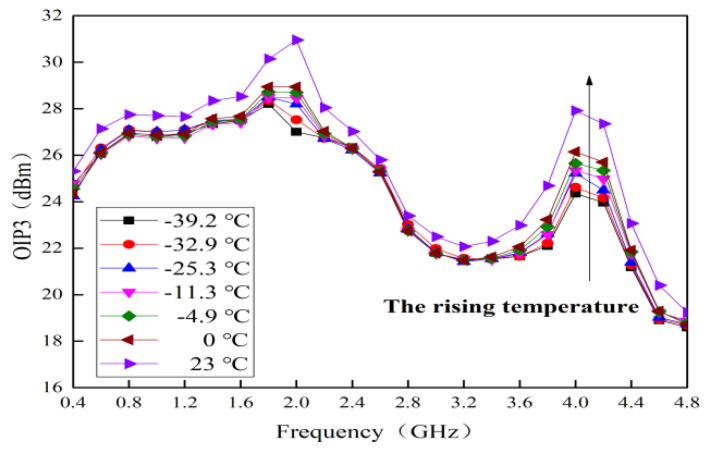
Measured curves of OIP3 with typical alpine temperatures.

**Figure 12 micromachines-13-01121-f012:**
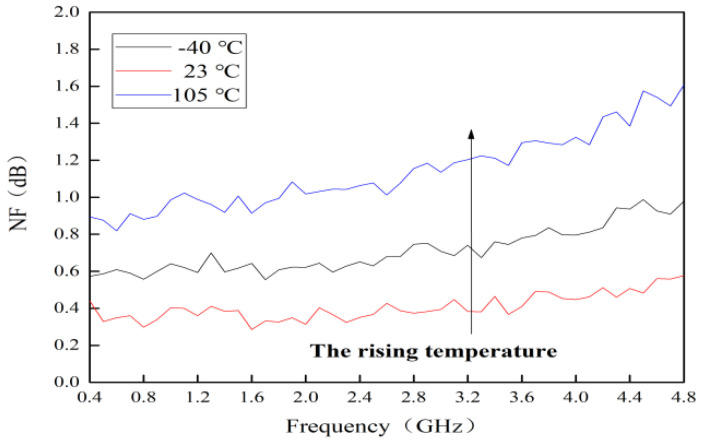
Measured curves of *NF* at different temperatures.

## Data Availability

The data presented in this study are available on request from the corresponding author.

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
