# Peer review of "Investigation on Temperature Behavior for a GaAs E-pHEMT MMIC LNA"

_micromachines, 2022, doi:10.3390/mi13071121_

Round 1

Reviewer 1 Report

The authors investigated how temperatures impacted the performance of a GaAs enhancement pseudomorphic high electron mobility transistor (pHEMT)-based high gain low-noise amplifier in this manuscript. Valuable experimental data were presented, however, some experimental details are missing. It can be reconsidered after significant revision. Some suggestions are as follows.

1.       Although the full names of abbreviations appeared in the abstract, they should be mentioned once in the main text to make this article readable.

2.       The stability K measured in this work should be defined.

3.       What literature on this topic has been found should be presented in the Introduction.

4.       Line 85 DFN should be explained. And I don’t get useful information from Fig2. Instead, showing the chips setup (with connections) may be better.

5.       Experimental details are missing. How were the temperatures achieved for this experiment? The ramping rate and dwell time should be considered.

6.       Line 111-113: Although the authors claimed that temperatures were set to simulate the plateau environment conditions, what about other factors such as pressure and humidity?

7.       The English of this article should be improved before the next submission. For instance, in Line 21, “engineer value” is not the correct term.

Reviewer 2 Report

This paper studied the temperature effect of GaAs LNA at 0.4-3.8 GHz under alpine condition. It is interesting to study the circuit performance in extreme environment. The performance of LNA are examined. I have some comments.

1. Fig. 1: the symbol of GaAs pHEMT should be modified. This is the nMOS symbol not HEMT.

2. section 4.3: What is k? please give the detail name. I think it is stability.

3. Fig. 11: why the 105 oC is used in NF test?

4. All the data in temperature below room temperature seems better. What should be consideration of the circuit under alpine condition?

5. Conclusions: Line 236: Meanwhile, this MMIC LNA still has "ow" noise performance. I think it is "low".

Round 2

Reviewer 1 Report

The manuscript has been significantly improved and I recommend it be published on Micromachines.